# Could Chronic Rhinosinusitis Increase the Risk of Ulcerative Colitis? A Nationwide Cohort Study

**DOI:** 10.3390/diagnostics12102344

**Published:** 2022-09-28

**Authors:** Il Hwan Lee, Seung-Su Ha, Gil Myeong Son, Hee Gyu Yang, Dong-Kyu Kim

**Affiliations:** 1Department of Otorhinolaryngology-Head and Neck Surgery, Chuncheon Sacred Heart Hospital, Hallym University College of Medicine, Chuncheon 24252, Korea; 2Institute of New Frontier Research, Division of Big Data and Artificial Intelligence, Chuncheon Sacred Heart Hospital, Hallym University College of Medicine, Chuncheon 24252, Korea

**Keywords:** rhinosinusitis, sinusitis, ulcerative colitis, cohort, innate immune

## Abstract

Chronic rhinosinusitis (CRS) is a common chronic inflammatory disease of the sinonasal mucosa with an inflammatory or infectious etiology. Inflammatory bowel disease (IBD) causes chronic intestinal inflammation. Thus, both diseases share innate immune and epithelial barrier dysfunctions of the mucosa. However, the association between sinusitis and IBD is not well-known. We aimed to determine the association between CRS and the risk for IBDs, such as Crohn’s disease (CD) and ulcerative colitis (UC). In this long-term retrospective cohort study, 15,175 patients with CRS and 30,350 patients without CRS (comparison group) were enrolled after 1:2 propensity score matching. The incidence rates of CD and UC were 0.22 and 0.51 (1000 person-years), respectively. The adjusted hazard ratio (HR) for developing CD and UC in CRS patients was 1.01 (95% confidence interval (CI), 0.66–1.54) and 1.72 (95% CI, 1.26–2.36), respectively. Additionally, in the subgroup analysis using the CRS phenotype, the adjusted HRs of UC were significantly increased in patients with CRS without nasal polyps (adjusted HR = 1.71; 95% CI, 1.24–2.35), but not in those with CRS with nasal polyps. CRS without nasal polyps is associated with an increased incidence of UC but not CD. Therefore, clinicians should pay attention to the early detection of UC when treating patients with CRS without nasal polyps.

## 1. Introduction

Chronic rhinosinusitis (CRS) is a heterogeneous chronic inflammatory condition of the sinonasal mucosa characterized by inflammation of the sinonasal passage, presenting with two or more sinonasal symptoms for more than 12 consecutive weeks [1]. It causes substantially impaired quality of life, reduces workplace productivity, and is related to significant direct and indirect economic costs. Clinically, CRS is classified into two phenotypes based on nasal endoscopic findings, namely, CRS with nasal polyps (CRSwNP) and CRS without nasal polyps (CRSsNP), which show different immunologic characteristics. Increasing evidence has revealed that CRSsNP is characterized by a predominant Th1 inflammatory response, whereas the immune response is skewed with Th2 inflammation in CRSwNP [2,3,4,5]. Additionally, CRS shows diminished epithelial barrier function, which induces the alteration of the host defense system [6,7].

Inflammatory bowel disease (IBD), represented by Crohn’s disease (CD) and ulcerative colitis (UC), is a chronic and remitting disorder responsible for causing inflammation of the gastrointestinal tract. Several epidemiologic studies have shown a higher prevalence of IBD in developed Western countries and an increasing incidence of IBD in various continents, including South America, Asia, Africa, and Eastern Europe [8,9,10]. IBD is also associated with an increased risk of a number of extraintestinal diseases [11,12]. Moreover, the recent hypothesis for the defective epithelial barrier has proposed mechanisms for the development of allergic and autoimmune diseases, including asthma, atopic dermatitis, allergic rhinitis, eosinophilic esophagitis, CRS, and IBD [13,14]. Among these, the association between CRS and IBD remains unclear, and previous reports also show discrepancies. Some earlier studies did not identify an association between CRS and IBD [15,16], whereas others showed a possible association between these diseases [17,18,19].

Therefore, this study aimed to investigate the risk of IBDs, such as CD and UC, in patients with CRS using a population-based long-term cohort dataset in South Korea.

## 2. Materials and Methods

### 2.1. Ethical Approval and Data Availability

The present study was approved by the Institutional Review Board (IRB) of Hallym Medical University, Chuncheon Sacred Hospital (No. 2021-08-006). The requirement for written informed consent was waived by the IRB because the South Korea National Health Insurance Service–National Sample Cohort database used in the study comprised de-identified secondary data. The datasets generated and/or analyzed in the present study are not publicly available because of the Korea National Health Insurance Service policies, yet are available from the corresponding author upon reasonable request.

### 2.2. Study Design and Population

This was a retrospective, nationwide propensity score-matched cohort study that used a dataset from the national health claims database. In this study, all disease diagnostic codes were identified using the Korean Classification of Disease, Fifth Edition modification of the International Classification of Disease and Related Health Problems, 10th revision (ICD-10). The CRS group included all patients who received inpatient or outpatient care for an initial diagnosis of CRS (J32 and J33) between January 2003 and December 2005. We established a washout period for 1 year (January 2002 and December 2002) to remove any potential pre-existing cases of CD (K50) and UC (K51). The operational definitions of the study endpoints were all-cause mortality or IBD events, such as CD and UC, during the follow-up period. Patients who experienced no events and were alive until 31 December 2013 were censored after this time point (Figure 1). To select the comparison group (non-CRS), we randomly identified propensity score-matched participants from the remaining cohort registered in the database as two participants without CRS for each patient with CRS. In this study, we selected the matched controls according to sex, age, residence, income level, and comorbidities. We categorized age, residence, income level, and comorbidities into three groups. Specifically, we adjusted the comorbidities using the Charlson Comorbidity Index (CCI), which is the most popular method used in studies based on claims datasets. It included 19 comorbidity conditions that, individually or in combination, predicted 1-year mortality risk. Patients diagnosed with CD and UC in 2002, or who died between 2003 and 2005, were excluded from the comparison group. Additionally, to improve the accuracy of the diagnosis of CRS, we operationally defined CRS in the present study as follows: (1) a diagnosis of rhinosinusitis more than twice within three months, which should be clinically persistent for more than 12 weeks; and (2) confirmation of chronicity verified by head and neck computed tomography (claims codes: HA401–HA416, HA441–HA443, HA451–HA453, HA461–HA463, or HA471–HA473). We also excluded patients aged <20 years, those who died between 2003 and 2005, and those diagnosed with IBD before the diagnosis of CRS.

### 2.3. Statistical Analyses

The incidence rates of CD and UC were compared between the CRS and comparison groups using person-years at risk, which were defined as the duration between the date of CRS diagnosis or 1 January 2003 (for the comparison group) and the patient’s respective endpoint. To identify whether CRS increased the risk of occurrence of specific diseases, we used Cox proportional hazards regression analysis to calculate the hazard ratio (HR) and 95% confidence interval (CI), adjusted for the other independent variables. During the follow-up period, the Kaplan–Meier method was used to calculate the specific disease-free survival rates among patients with CRS. All statistical analyses were performed using R version 4.0.5 (URL https://www.R-project.org/, accessed on 1 March 2021), with a significance level of *p* = 0.05.

## 3. Results

A total of 30,350 participants without CRS and 15,175 patients with CRS were enrolled in this study during the 10-year follow-up period. The characteristics of the study population in each group are shown in Table 1. The CRS population consisted of 6108 (40.3%) men and 9067 (59.7%) women. We analyzed the balance plot between the comparison and CRS groups to confirm the effectiveness of propensity score matching. We found similar distributions between the two groups, indicating that each variable was appropriately matched (Appendix A).

### 3.1. Effect of Chronic Rhinosinusitis on Inflammatory Bowel Diseases

To compare the risk of subsequent development of IBD, in the present study, 297,761.1 person-years in the non-CRS group and 143,933.7 person-years in the CRS group were evaluated for CD events. Meanwhile, 297,675.4 person-years in the non-CRS group and 143,654.1 person-years in the CRS group were evaluated for UC events. The overall incidences of CD and UC were evaluated at 0.22 and 0.51 per 1000 person-years in the CRS group, respectively (Table 2). We also present a description of time to event and censored data in Table 3. The HRs for the development of CD and UC using univariate and multivariate Cox regression models are shown in Table 2. After adjusting for sex, age, residence, income level, and comorbidities, we found that CRS was significantly associated with the development of UC (adjusted HR = 1.72; 95% CI, 1.26–2.36); however, there was no significant difference in the subsequent development of CD between the two diseases. The Kaplan–Meier survival analysis revealed that patients in the CRS group presented a more frequent incidence of ulcerative colitis events than those in the control group (Figure 2).

### 3.2. Chronic Rhinosinusitis with Nasal Polyp and Ulcerative Colitis

Next, we further analyzed the HR of CD and UC events according to the CRS phenotype (Table 4). We found that patients with CRSsNP showed a significant association with the development of ulcerative colitis, with an adjusted HR of 1.71 (95% CI, 1.24–2.36), but we could not detect a significant difference in patients with CRSwNP. However, the CD event showed no significant difference in association with the CRS phenotype. The risk of UC after CRSsNP development was similar and constant according to the follow-up period (Table 5 and Figure 3). This indicates that the association between CRSsNP and UC may not be a temporal incident.

## 4. Discussion

In this study, we investigated whether CRS increases the risk of IBD events. For this analysis, we selected participants who were matched for sociodemographic factors from a nationwide 11-year longitudinal cohort database of 1,025,340 South Korean patients. In South Korea, the incidence and prevalence of IBD is relatively low compared to Western countries (Table 6 and Figure 4). However, to the best of our knowledge, the present study is the first to investigate the development of IBD in patients with CRS using a nationwide population-based dataset. We found a significant difference between the CRS and comparison groups in the number of patients who developed UC, specifically in patients with CRSsNP, but not in patients with CRSwNP. After adjusting for sociodemographic characteristics and comorbidities, patients with CRSsNP showed 1.72 times higher risk of developing UC than those without CRSsNP. However, we also observed that patients with CRS exhibited no significantly increased risk of CD, regardless of its phenotype.

Although the pathophysiology of CRS is multifactorial, CRS may share some immunologic pathophysiology with IBD because the sinonasal and intestinal surfaces are lined by epithelial cells that interact with environmental factors and immune cells. Thus, immune and epithelial barrier dysfunctions, including enhanced epithelial expression of Toll-like receptors, nucleotide-binding oligomerization domain-like receptors, decreased defensing, and tight junction dysfunction, are detected in the nasal and gut mucosa, respectively [20,21,22,23,24]. For these reasons, to date, several researchers have thoroughly investigated the association between sinusitis and IBD. In real-world studies, patients with IBD often showed more nasal and sinus symptoms than the control group [12,17,18]. However, some large-scale, retrospective studies showed that there was no significant evidence for the association between IBD and the subsequent development of CRS [15,25].

Recently, a study using a nationwide dataset revealed that patients with IBD were at greater risk of developing CRS later in life than those without IBD, especially the phenotype of UC [17]. Similar to this study, we found an increased risk of subsequent UC development in patients with CRSsNP, but not in patients with CRSwNP. Additionally, another recent study investigating the bidirectional development between sinusitis and IBD reported that the duration of IBD, UC, steroid exposure, and younger age of IBD diagnosis were associated with subsequent sinusitis in patients with UC, whereas steroid exposure and duration of sinusitis were significantly associated with subsequent IBD in patients with sinusitis [25]. For these reasons, we hypothesize that UC and CRSsNP have been intrinsically associated with a network of deregulated inflammatory mediators, such as the interleukin (IL)-33/ST2 pathway [26,27,28,29]. IL-33 is one of the major innate cytokines involved in the pathophysiology of CRS in Asian populations. It causes long-standing inflammatory stress on the sinonasal epithelium and is highly expressed in the sinonasal tissues of patients with CRS [29,30]. IL-33 is also associated with UC through aberrant IL-33/ST2 signaling and is correlated with the clinical activity of IBD in murine models and human studies [31,32,33]. Specifically, several studies have shown higher IL-33 expression in patients with UC compared with healthy controls [34,35]. Moreover, blockade of IL-33/ST2 signaling can alleviate active UC status in a murine model [36]. Additionally, one prior study revealed that sinusitis-derived *Staphylococcal enterotoxin B* may be associated with the development of UC [19]. Moreover, similar with CRSwNP, some patients with CRSsNP are increased the expression of IL-4, IL-5, *Staphylococcus aureus enterotoxin-specific IgE*, and elevated numbers of eosinophils in both the blood and nasal mucosa [37]. Additionally, the presentation of type 2 inflammation in patients with CRSsNP is usually associated with a worse clinical outcome [38]. Interestingly, we detected a constant HR for developing UC during the 11-year follow-up in patients with CRSsNP. Collectively, these findings suggest that both diseases may be intrinsically associated with an elaborate inflammatory network, although we could not determine the exact mechanism.

This study had several unique strengths. First, our dataset based on a nationwide population has been used in multiple recently published studies, and this dataset has already proven the reliability of the prevalence of major diseases [39,40]. Second, these findings have meaningful clinical implications. That is, the simultaneous extra-sinonasal examination when physicians treat patients with CRS may be helpful for the early detection and timely treatment of IBD. Based on our findings, clinicians should consider the association between CRS and UC and assess for gastrointestinal manifestations during their practice.

However, our study also has some limitations. First, the diagnosis of CRS events was mainly dependent on ICD-10 diagnostic codes, which may be less accurate than diagnoses based on medical chart data, as these usually include medical history and nasal examination results. Thus, a misclassification bias may have occurred in this study. However, to minimize this issue, we added the inclusion criteria for performing head and neck computed tomography during the selection of patients with CRS. Second, we could not consider IBD-specific factors, such as medication exposure, prolonged chronic inflammation, and bowel obstruction. Therefore, we did not evaluate how CRS affects the severity of IBD. Additionally, since we could not adjust CRS treatment during the follow-up period, the discrepancy of CRS treatment could be played as one of the confounding factors. Third, our database provides categorized age data (<45, 45–64, and >64 years). Therefore, we could not match the two groups according to the actual age distribution, and our findings may have had some residual bias within the categories. Fourth, our dataset did not include information about disease duration (onset history) and severity of CRS (Lund-Mackay score); therefore, we were unable to investigate whether the duration or severity of CRS may have a differential effect on the IBD risk. Fifth, we could not consider the effect of CRS medication, such as antibiotics and steroids, because we could not obtain data on medication compliance for each patient. However, to overcome these limitations, we selected a comparison group (non-CRS participants) who were socio-demographically matched with patients with CRS using the propensity score matching method. Finally, due to the retrospective cohort design of the present study, we could not determine whether our findings regarding the association between CRS and IBD have a real underlying common pathology or are just a temporal incident. Further studies are required to confirm this hypothesis.

In conclusion, the present study examined the association between CRS and the risk of IBD events after adjusting for clinical and demographic factors. We identified an increased risk of UC events in patients with CRSsNP; however, no significant association was detected between CD and CRS, regardless of its phenotype. Therefore, clinicians should be aware of the potential development of UC in patients with CRSsNP and recommend gastrointestinal examination to ensure early detection.

## Figures and Tables

**Figure 1 diagnostics-12-02344-f001:**
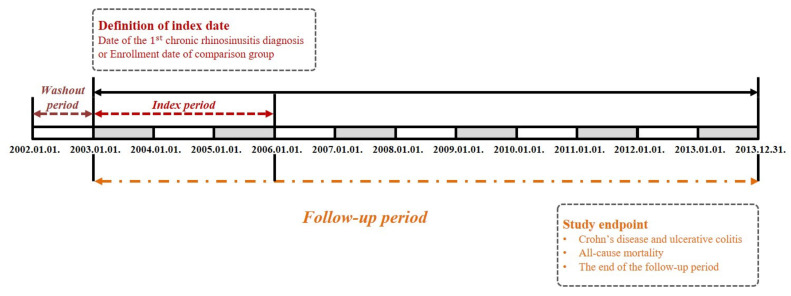
Schematic flow of the study design.

**Figure 2 diagnostics-12-02344-f002:**
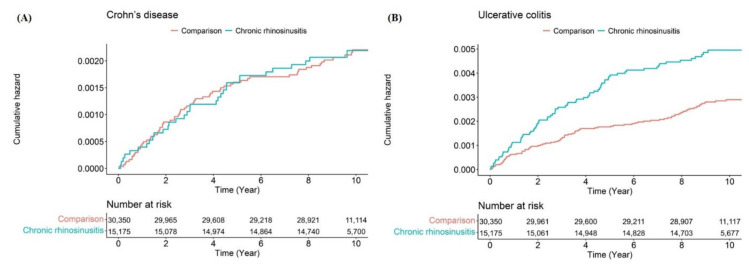
Cumulative hazard plot of (**A**) Crohn’s disease and (**B**) ulcerative colitis between the chronic rhinosinusitis and comparison groups.

**Figure 3 diagnostics-12-02344-f003:**
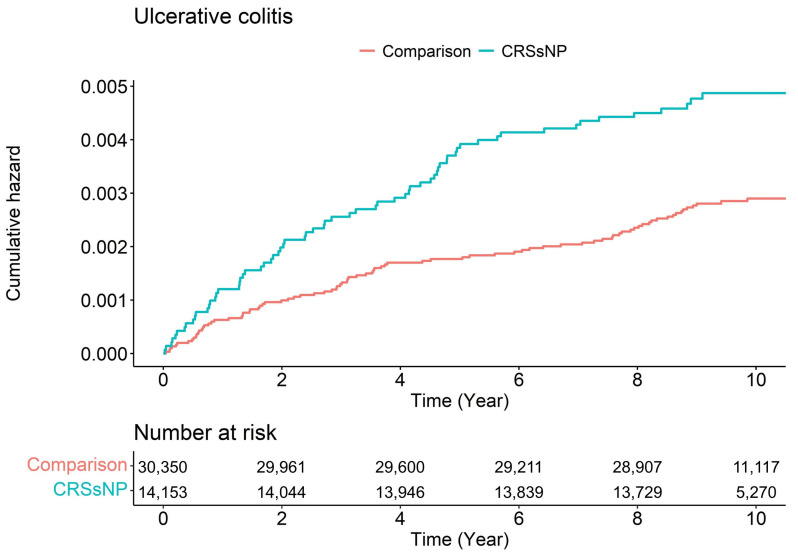
Hazard ratios for incident ulcerative colitis associated with the development of chronic rhinosinusitis with nasal polyp by time since the diagnosis of chronic rhinosinusitis without nasal polyp.

**Figure 4 diagnostics-12-02344-f004:**
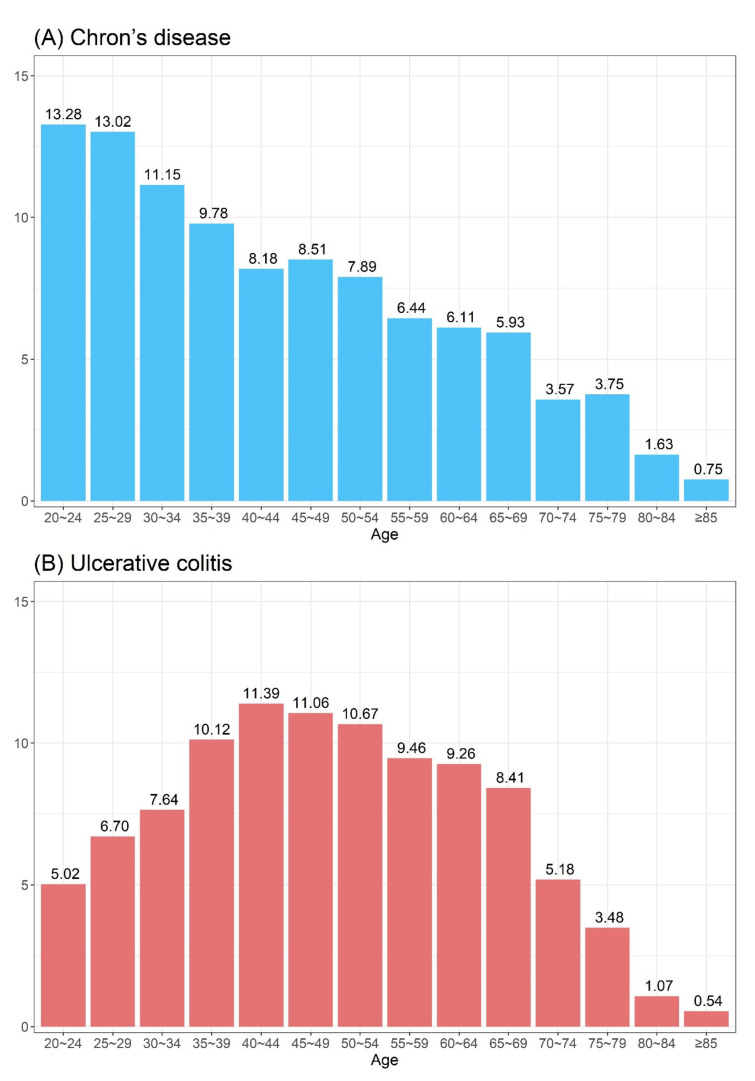
Age distribution of patients with (**A**) Crohn’s disease and (**B**) ulcerative colitis.

**Table 1 diagnostics-12-02344-t001:** Detailed characteristics of each cohort dataset.

Variables	Comparison (n = 30,350)	Chronic Rhinosinusitis (n = 15,175)	*p*-Value
Sex			1.000
Male	12,216 (40.3%)	6108 (40.3%)	
Female	18,134 (59.7%)	9067 (59.7%)	
Age (years)			1.000
<45	18,514 (61.0%)	9257 (61.0%)	
45–64	9322 (30.7%)	4661 (30.7%)	
>64	2514 (8.3%)	1257 (8.3%)	
Residence			1.000
Seoul (largest city)	8106 (26.7%)	4053 (26.7%)	
Other metropolitan	8004 (26.4%)	4002 (26.4%)	
Other areas	14,240 (46.9%)	7120 (46.9%)	
Income level			1.000
Low (0–30%)	5620 (18.5%)	2810 (18.5%)	
Middle (30–70%)	11,294 (37.2%)	5647 (37.2%)	
High (70–100%)	13,436 (44.3%)	6718 (44.3%)	
CCI			1.000
0	18,964 (62.5%)	9482 (62.5%)	
1	6876 (22.7%)	3438 (22.7%)	
≥2	4510 (14.9%)	2255 (14.9%)	

CCI, Charlson Comorbidity Index; Comparison, participants without CRS; second area, other metropolitan cities; Seoul, the largest metropolitan area; third area, other areas.

**Table 2 diagnostics-12-02344-t002:** The incidence rate (1000 person-years) and HR (95% CI) of incident inflammatory bowel diseases such as Crohn’s disease and ulcerative colitis associated with patients with CRS.

Variables	N	Case	Person-Years	Incidence Rate	Crude HR (95% CI)	Adjusted HR (95% CI)
Crohn’s disease						
Comparison	30,350	64	297,761.1	0.21	1.00 (ref)	1.00 (ref)
CRS	15,175	32	143,933.7	0.22	1.01 (0.66–1.54)	1.01 (0.66–1.54)
Ulcerative colitis						
Comparison	30,350	85	297,675.4	0.29	1.00 (ref)	1.00 (ref)
CRS	15,175	73	143,654.1	0.51	1.74 (1.27–2.38) ***	1.72 (1.26–2.36) ***

*** *p* < 0.001. CI, confidence interval; comparison, participants without CRS; CRS, chronic rhinosinusitis; HR, hazard ratio.

**Table 3 diagnostics-12-02344-t003:** Description of time to event and censored data.

	The Number of CD Events	The Number of UC Events
Event	96	158
Comparison	64	85
CRS	32	73
Total censored (No event)	45,429	45,367
Comparison	30,286	30,265
CRS	15,143	15,102
Termination of study	43,229	43,169
Comparison	28,582	28,562
CRS	14,647	14,607
Loss to follow up/Drop-out	2200	2198
Comparison	1704	1703
CRS	496	495

Comparison, participants without CRS. CD, Crohn’s disease; UC, ulcerative colitis; CRS, chronic rhinosinusitis.

**Table 4 diagnostics-12-02344-t004:** The incidence rate (1000 person-years) and HR (95% CI) of Crohn’s disease and ulcerative colitis according to the phenotype of chronic rhinosinusitis.

Variables	N	Case	Person-Years	Incidence Rate	Crude HR (95% CI)	Adjusted HR (95% CI)
Crohn’s disease						
Comparison	30,350	64	297,761.1	0.21	1.00 (ref)	1.00 (ref)
CRSsNP	14,153	32	134,205.9	0.24	1.08 (0.71–1.66)	1.08 (0.71–1.65)
CRSwNP	1022	0	9727.8	0	0.00 (0–Inf)	0.00 (0–Inf)
Ulcerative colitis						
Comparison	30,350	64	297,761.1	0.21	1.00 (ref)	1.00 (ref)
CRSsNP	14,153	67	133,960.2	0.50	1.71 (1.24–2.36) **	1.71 (1.24–2.35) **
CRSwNP	1022	6	9693.8	0.62	2.14 (0.93–4.89)	1.97 (0.86–4.51)

** *p* < 0.010. CI, confidence interval; comparison, participants without CRS; CRSsNP, chronic rhinosinusitis without nasal polyp; CRSwNP, chronic rhinosinusitis with nasal polyp; HR, hazard ratio.

**Table 5 diagnostics-12-02344-t005:** Hazard ratios for incident ulcerative colitis associated with CRSsNP development by time since CRSsNP diagnosis.

Time (Year)	Number of Ulcerative Colitis		Adjusted HR (95% CI)
	Comparison	CRSsNP	
1	19	17	1.94 (1.01–3.73) *
2	30	28	2.01 (1.20–3.37) **
3	39	36	1.99 (1.26–3.12) **
4	51	41	1.73 (1.14–2.60) **
5	53	54	2.18 (1.49–3.19) ***
6	57	58	2.18 (1.51–3.14) ***
7	61	60	2.10 (1.47–3.00) ***
8	70	63	1.92 (1.36–2.70) ***
9	83	66	1.72 (1.24–2.37) **
10	85	67	1.71 (1.24–2.35) **
11	85	67	1.71 (1.24–2.35) **

* *p* < 0.05, ** *p* < 0.010, and *** *p* < 0.001. CI, confidence interval; CRSsNP, chronic rhinosinusitis without nasal polyp; HR, hazard ratio.

**Table 6 diagnostics-12-02344-t006:** Prevalence of Crohn’s disease and ulcerative colitis according to the year (from 2002 to 2013).

Year	Chron’s Disease (Patients Aged over Twenty)	Ulcerative Colitis (Patients Aged over Twenty)
2002	567 (0.08%)	491 (0.07%)
2003	477 (0.06%)	642 (0.09%)
2004	447 (0.06%)	557 (0.07%)
2005	376 (0.05%)	564 (0.07%)
2006	303 (0.04%)	545 (0.07%)
2007	220 (0.03%)	546 (0.07%)
2008	226 (0.03%)	573 (0.08%)
2009	189 (0.02%)	576 (0.08%)
2010	224 (0.03%)	589 (0.08%)
2011	283 (0.04%)	647 (0.08%)
2012	264 (0.03%)	617 (0.08%)
2013	288 (0.04%)	642 (0.08%)

## Data Availability

The authors confirm that the data supporting the findings of this study are available within the article.

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
