# Peer review of "Could Chronic Rhinosinusitis Increase the Risk of Ulcerative Colitis? A Nationwide Cohort Study"

_diagnostics, 2022, doi:10.3390/diagnostics12102344_

Round 1
Reviewer 1 Report
I appreciate the opportunity to review the manuscript for publication in MDPI Diagnosis.
The authors aim to determine possible association between CRS and the risk for IBDs, such as Crohn’s disease (CD) and ulcerative colitis (UC). It worth publish that the adjusted hazard ratio (HR) for developing UC in CRS patients was 1.72 (95% CI, 1.26–2.36). The CRSsNP subgroup showed increased the adjusted HRs of UC but not in those with CRSwNP. This is consistent with endotype classification of CRS, with the CRSsNP patients being more deviated toward Type 1 inflammation.
I feel that the topics are interesting and the manuscript is well organized. I have following comments.
L79: “income level, and comorbidities.”
Please explain comorbid diseases more in detail. The history of medication is another important factor.
L89: “those diagnosed with cholesteatoma”
What are the specific reasons for enrollment of cholesteatoma?
Table 2. The incidence rate (1,000 person-years) and HR (95% CI) of incident inflammatory bowel diseases such as Crohn's disease and ulcerative colitis associated with patients with CRS.
I am surprised that incident rates of IBD were quite low than expected as compared those in our country. The authors should disclose absolute average numbers and age distribution of IBD patients in Korea, whether they suffer CRS or not.
The authors should clarify the treatment of CRS in the observed periods.
Figure 3. Hazard ratios for incident ulcerative colitis associated with the development of chronic rhinosinusitis with nasal polyp by time since the diagnosis of chronic rhinosinusitis without nasal polyp.
It is recommended that the figure is modified to the cumulative hazard plot as Figure 2 for better understanding.
In the discussion, the authors also had better focus on the difference in endotypes between CRSsNP and CRSwNP in Asian population. Possible relation with systemic eosionophilia might be involved or not.
Author Response
Point 1: Please explain comorbid diseases more in detail. The history of medication is another important factor.
Response 1: Firstly, thank you for your kind review. The Charlson Comorbidity Index was first developed in 1987 by Mary Charlson and colleagues as a weighted index to predict risk of death within 1 year of hospitalization for patients with specific comorbid conditions. Nineteen conditions were included in the index. We added more description as follows: “It included 19 comorbidity conditions that, individually or in combination, predicted 1-year mortality risk”.
Point 2: “those diagnosed with cholesteatoma” What are the specific reasons for enrollment of cholesteatoma?
Response 2: This is our mistake. We’re very sorry about confusing to the reviewers. We modified “cholesteatoma” to “IBD”.
Point 3: Table 2. The incidence rate (1,000 person-years) and HR (95% CI) of incident inflammatory bowel diseases such as Crohn's disease and ulcerative colitis associated with patients with CRS. I am surprised that incident rates of IBD were quite low than expected as compared those in our country. The authors should disclose absolute average numbers and age distribution of IBD patients in Korea, whether they suffer CRS or not.
Response 3: We totally agreed with the reviewer’s comments; however, in South Korea, the incidence and prevalence of IBD is relatively low compared to Western countries. We added the absolute average numbers of IBD patients and its age distribution in our database as Table 6 and Figure 4.
Point 4: The authors should clarify the treatment of CRS in the observed periods.
Response 4: Unfortunately, we could not adjust CRS treatment during the follow-up period because each patient showed very heterogeneous treatment course. So, we added this issue as limitations on the section of Discussion, as follows: “Additionally, since we could not adjust CRS treatment during the follow-up period, the discrepancy of CRS treatment could be played as one of the confounding factors”.
Point 5: Figure 3. Hazard ratios for incident ulcerative colitis associated with the development of chronic rhinosinusitis with nasal polyp by time since the diagnosis of chronic rhinosinusitis without nasal polyp. It is recommended that the figure is modified to the cumulative hazard plot as Figure 2 for better understanding.
Response 5: We modified Figure 3 as you recommended.
Point 6: In the discussion, the authors also had better focus on the difference in endotypes between CRSsNP and CRSwNP in Asian population. Possible relation with systemic eosionophilia might be involved or not.
Response 6: We totally agreed with the reviewer’s comments. Thus, we added more description and citation for endotype heterogeneity in the Discussion.

Reviewer 2 Report
The methodology of study is good and flawless.
The result is clearly presented.
The discussion is well-written with adequate references.
Congratulation for your excellent work.
Author Response
Firstly, thank you for your kind review.